# Lactic Acid Production by *Lactiplantibacillus plantarum* AC 11S—Kinetics and Modeling

**DOI:** 10.3390/microorganisms12040739

**Published:** 2024-04-04

**Authors:** Petya Popova-Krumova, Svetla Danova, Nikoleta Atanasova, Dragomir Yankov

**Affiliations:** 1Institute of Chemical Engineering, Bulgarian Academy of Sciences, 103 Acad. G. Bontchev Str., 1113 Sofia, Bulgaria; p.krumova@iche.bas.bg; 2The Stephan Angeloff Institute of Microbiology, Bulgarian Academy of Sciences, 26 Acad. G. Bontchev Str., 1113 Sofia, Bulgaria; stdanova@yahoo.com (S.D.); nikoletaatanasova21@gmail.com (N.A.)

**Keywords:** lactic acid, *Lactiplantibacillus plantarum*, lactose, growth, mathematical modeling

## Abstract

Lactic acid is a versatile chemical with wide application in many industries. It can be produced by the fermentation of different sugars by various *lactobacilli* and investigations on lactic acid production from different substrates and by different strains are still in progress. The present study aimed to study lactic acid production from lactose by *Lactiplantibacillus plantarum* AC 11S and to choose a mathematical model describing in the best way the experimental data obtained. The influence of initial substrate concentration was investigated, and optimal pH and temperature were determined. An unstructured mathematical model was developed comprising equations for bacterial growth, substrate consumption, and product formation. The model was solved with different terms for specific growth rates considering substrate and/or product inhibition. The best bacterial growth and lactic acid production were achieved at pH = 6.5 and 30 °C. Production of lactic acid was mainly growth-associated, and at initial substrate concentration over 15 g/L, a considerable product inhibition was observed. The parameters of different models were determined and compared. The modified Gompertz equation gave the best fit when solving only the equation for biomass growth at different initial substrate concentrations. Solving the entire set of differential equations for bacterial growth, substrate consumption, and product formation, the best results were obtained when using a variant of the logistic equation for biomass growth. This variant included a term for product inhibition and described in the best way all experimental data. Solving the model for different biomass concentrations showed that an increase in biomass led to a shorter lag phase and the stationary phase was reached faster. The results obtained, optimum conditions and the kinetic model, are good bases for studying pH-controlled fermentation, as well as a continuous process.

## 1. Introduction

Lactic acid bacteria (LAB) are a multitudinous microbial group of microorganisms with diverse beneficial applications in the food and pharmaceutical industry. LAB share the same characteristics of metabolism and physiology with strain-specific functionality and potential. One of their main common features is the production of lactic acid as an end-product of the fermentation of various carbohydrates. From a taxonomic point of view, they comprise a large variety of Gram-positive, catalase-negative, and anaerobic, non-sporulation bacteria belonging to the *Lactobacillaceae* family, with different G+C *Firmicutes* groups. They are producers of organic acids, polyols, bacteriocins, exopolysaccharides (EPSs), and aromatic compounds, among others [1]. Several LAB species are widely accepted as probiotics with functional characteristics and beneficial healthy effects based on their metabolic activity. 

*Lactiplantibacillus plantarum* can be found in various ecological niches, such as in dairy products, fermented vegetables and cereals, meat, fish, silage, wine, and gastrointestinal, urogenital, and vaginal tracts [2]. It is one of the most exploited LAB for producing various postmetabolites. *L. plantarum* is well known for its versatility in substrate utilization [3]. Various *L. plantarum* strains are capable of utilizing different monosaccharides, either hexoses or pentoses [4], as well as different lignocellulosic hydrolysates from rice straw [5], corncobs [6], sugarcane bagasse [7], *Opuntia ficus indica* waste [8], and algal biomass [3], among others. These species are considered the most advantageous among LAB in the production of lactic acid (LA) from lactose and they are also capable of utilizing other nutrients presented in whey [9]. 

*L. plantarum* has a larger genome than other LAB species (size of about 3.3 Mbp and more than 3000 genes), which indicates its strong adaptability, high versatility, enormous diversity in phenotypic properties, metabolic capacity, and industrial applications [10]. According to Zheng et al. [11], the *L. plantarum* group represents the evolutionary link between heterofermentative and homofermentative LAB. Though *L. plantarum* is phylogenetically related to heterofermentative *lactobacilli*, it shares different metabolic features with homofermentative *lactobacilli*. The main difference between homo- and heterofermentative lactobacilli is the existing difference in the metabolic pathways leading to variation in the main end-product of the fermentation. While homofermentative *lactobacilli* produce lactic acid as the sole product, the heterofermentative LAB produce a mixture of lactic acid, ethanol, acetic acid, and carbon dioxide during fermentation. Other differences also exist in the ability of various LAB to decompose macromolecular substances (polysaccharides and proteins), as well as in the production of bacteriocins, short-chain fatty acids, amines, etc. 

LAB have been used by humans for centuries. Several metabolic end-products of lactic acid fermentation have practical applications. Lactic acid is a very useful chemical with numerous applications in pharmaceuticals, foods, cosmetics, and other industries. In recent years, the use of LAB in drug delivery systems, fermented foods, and beverages, as well as probiotics and vaccines, has gained more and more interest. *L. plantarum* has been widely used as a model species for ecological, genetic, and metabolic studies in *lactobacilli*. The major commercial use of *L. plantarum* is as a starter culture for various food fermentations and in recent years as a probiotic culture [11].

*L. plantarum* is considered to belong to the group of facultatively heterofermentative *lactobacilli*. Via the Embden–Meyerhof–Parnas pathway, hexoses are almost completely converted to lactic acid. On the other hand, pentoses are converted to acetic and lactic acids via the 6-phosphogluconate/phosphoketolase pathway by the induction of phosphoketolase [12]. *L. plantarum*, like other LAB, possesses specific nutritional requirements and thus, a complex growth media, which must contain peptides, vitamins, and nucleic, fatty, and amino acids, besides the carbon source is needed. Another particular characteristic of the process of lactic acid production is the end-product inhibition. Due to the lactic acid accumulation, the fast acidification of the fermentation medium leads to low substrate utilization, low conversion, and low final product concentration.

In the process of a comprehensive investigation of lactic acid production, kinetic studies are an indispensable step. A proper mathematical model is very useful in understanding, describing, evaluating, and predicting the fermentation process. Mathematical models help disclose the relationship between cell growth, substrate consumption, and product formation and can be used for control optimization and scale-up of a fermentation process. Mathematical models have been classified according to different criteria [13]. One of these criteria distinguishes between structured and unstructured mathematical models. Structured models are more accurate and take into account cell functions, structure, and composition. Unstructured models, on the other hand, treat the cells as a black box, a sole component without involving any cells’ physiological characteristics. Although the structured models are more detailed and precise, providing a better understanding of the system, unstructured models are widely used, because of their applicability and simplicity in describing bacterial kinetics. Different models have been used in the modeling of lactic acid production from various substrates. Some of them are summarized in the works of Zacharof & Lovitt [13], Bouguettoucha et al. [14], and Gordeev et al. [15].

The present work aims to investigate *Lactiplantibacillus plantarum* strain AC 11S, as a promising producer of lactic acid from lactose, to determine the optimum conditions for growth and production, and to compare some kinetics models describing the fermentation dynamics.

## 2. Materials and Methods

### 2.1. Strain, Growth Media, and Culture Conditions

*L. plantarum* AC 11S was isolated from a homemade white brined cheese [16]. It was cultured in de Man, Rogosa, and Sharpe broth (MRS Merck, Darmstadt, Germany) and stored at −20 °C in MRS broth supplemented with glycerol 20% *v*/*v*. Before the experiments, the strain was pre-cultured twice in modified MRS broth with lactose. The strain was cultured in growth medium containing 11 g/L lactose monohydrate, 5.5 g/L yeast extract, 12.5 g/L peptone from casein, 10 g/L sodium acetate, 0.25 g/L KH_2_PO_4_, 0.25 g/L K_2_HPO_4_, 0.1 g/L MgSO_4_·7H_2_O, 0.05 g/L MnSO_4_·7H_2_O, 0.05 g/L Fe_2_(SO_4_)_3_. All chemicals were p.a. grade (Fluka, Darmstadt, Germany). The inoculum (10% *v*/*v*) was prepared from overnight culture in MRS broth with lactose monohydrate (11 g/L). All experiments were carried out in flasks (100 mL growth medium in 300 mL Erlenmeyer flasks at 30 °C with an initial pH of 6.5) using a WiseCube^®^ WIS30 shaking incubator (Witeg Labortechnik GmbH, Wertheim, Germany).

### 2.2. The LAB Species Identification 

The strain AC 11S was identified as *L. plantarum* by classical phenotypic methods including Gram staining, oxidase (kit HiMedia, Mumbai, India), catalase tests, and a carbohydrate fermentation test with 49 carbon sources (using API 50 CHL, bioMérieux, Marcy l’Etoile, France). The species affiliation was confirmed by Multiplex PCR amplification with primers targeting the *recA* gene, according to Torriani et al. [17]. The PCR analysis was performed on a PCR thermocycler (BioRad, laboratories, Inc. group Hercules, CA, USA), using a ready to use PCR mix (IllistraTM PuRe TaqTM Ready To GoTM PCR beads; Amersham Biosciences, Amersham, UK). The target DNA was isolated from an overnight LAB culture of the strain AC 11S using the Gene Matrix Bacterial and Yeast Genomic DNA Purification Kit (EURx Ltd., Gdańsk, Poland), following the manufacturer’s instructions. The PCR reaction conditions were as previously described [18]. 

### 2.3. In Vitro Assessment of Acid Tolerance of L. plantarum AC 11S

An in vitro test, in 96-well U-bottomed, polystyrene microtiter plates (Corning, NY, USA) with a final volume of 150 µL, was carried out. The cells from 10 mL overnight culture were harvested by centrifugation (centrifuge 9000× *g*, Hermle Labortechnik GmbH, Wehingen, Germany) and washed twice with sterile saline. Washed cells re-suspended in 10 mL were adjusted to 6.0 × 10^8^ CFU/mL (2× McFarland). The cells from 5 mL were harvested and re-suspended in 5 mL simulated gastric fluid with a low pH of 1.5, 0.8% NaCl (*w*/*v*), and 3 mg/mL pepsin from porcine gastric mucosa (Sigma-Aldrich, Saint Louis, MO, USA). The resting 5 mL was used as a control re-suspended in sterile saline of pH 6.5. They were incubated for 3 h at 30 °C. After treatment, they were washed 2 times, with sterile saline. Following the treated culture, the control was used as an inoculum (10% *v*/*v*) in modified MRS broth (with lactose 20 g/L) in 96-well microplates. Viability, after treatment, was monitored spectrophotometrically (OD 600 nm) during 24 h cultivation at 30 °C using an Elisa Plate Reader (INNO, Seongnam-si, Republic of Korea). The results are presented as a means of triplicate.

### 2.4. Analytical Procedures

For biomass determination, optical density (OD) was measured at 620 nm (UV-VIS spectrophotometer Milton Roy 401, Rochester, NY, USA). From OD data, by using a previously prepared calibration curve, the biomass concentration was calculated. An HPLC system composed of a Knauer Smartline-100 pump, a Perkin-Elmer LC-25RI refractometric detector, and data processing software Eurochrome v. 3.05 (Knauer, Berlin, Germany) were used for lactic acid and lactose measurements. An Aminex HPX-87H (Bio-Rad, Hercules, CA, USA) column was used. A 0.005 M solution of H_2_SO_4_ at a flow rate of 0.6 mL/min was used as the mobile phase. For standard solutions preparation, pure (98% mass, Sigma, Darmstadt, Germany), crystalline L (+)-lactic acid was used. All measurements were taken in triplicate. The pH was measured using a pH meter HI2211 (HANNA instruments, Bedfordshire, UK). Correction of pH to pH = 1.5 was achieved with hydrochloric acid.

### 2.5. Fermentation

All experiments for lactic acid production were carried out in MRS broth with lactose as a carbon source in 300 mL flasks at 30 °C and a 6.5 initial pH of the medium (except those for optimum pH and temperature determination).

### 2.6. Modeling

The main goal in a fermentation process is complete substrate utilization, leading to a maximum product yield. The development of a proper mathematical model and correct determination of model parameters will permit us to conduct the fermentation in its optimal conditions. The mathematical model should contain equations describing cell growth, substrate consumption, and product formation and their interrelationship. In an unstructured mathematical model, only the total cell concentration is considered. In general, the bacterial growth rate can be described as follows: (1)$$\frac{dX}{dt}=\mu X-k_{d}X$$
where *X* is the biomass concentration, g/L; *t* is time, h; *μ* is the specific growth rate, h^−1^; and *k_d_* is the specific cell death rate, h^−1^. In many cases, cell death was not observed or *µ* >> *k_d_* and the second term can be omitted.

The simplest model representing the specific growth rate as a function of substrate concentration is the Monod equation:(2)$$\mu =\mu_{max}\left(\frac{S}{K_{S}+S}\right)$$
where *μ_max_* is the maximum specific growth rate, h^−1^; *S* is the initial substrate concentration, g/L; and *K_S_* is the substrate saturation constant, g/L. *K_S_* represents substrate concentration at which the growth rate is equal to half of the maximum growth rate. The Monod equation describes the increase in cell concentration with time and considers limiting substrate concentration [13,14,15]. It is widely accepted that lactic acid fermentation is limited by the substrate and inhibited by the product. Many authors insert different terms in the Monod equation to take inhibition effects into account. Some of them are presented in Table 1.

Other authors have used different sigmoidal models for the description of microbial growth. The most utilized are the logistic equation and the modified Gompertz model:(3)$$\ln\left(\frac{X}{X_{0}}\right)=\frac{a}{\left[1-exp\left(b-ct\right)\right]}$$
(4)$$\ln\left(\frac{X}{X_{0}}\right)=a\;exp\left[-exp\left(b-ct\right)\right]$$

These models do not include the consumption of the substrate and describe only the number of cells.

With the aim to give a biological meaning to the parameters *a*, *b*, and *c*, Zwietering [26] reparametrized the above equations to
(5)$$X\left(t\right)=X_{0}+\frac{A}{1+exp\left[\frac{4\mu_{max}}{A}\left(\lambda -t\right)+2\right]}$$
(6)$$X\left(t\right)=X_{0}+Aexp\left\{-exp\left[\frac{\mu_{max}e}{A}\left(\lambda -t\right)\right]+1\right\}$$

In these equations, *X*_0_ is the initial biomass concentration, g/L; *t* is time, h; *A* is the asymptotic level of biomass, g/L; *λ* is the duration of the lag phase, h; and *µ_max_* is the maximum specific growth rate, h^−1^.

The models listed in Table 1, as well as the logistic and Gompertz equation, were chosen in view to include different types of inhibition.

The Luedeking–Piret equation [27] is the most used for the description of lactic acid formation. It takes into account the fact that the rate of product accumulation *dP*/*dt* depends on bacterial growth *dX*/*dt*, as well as on the bacterial density *X*.
(7)$$\frac{dP}{dt}=\alpha \frac{dX}{dt}+\beta X\;$$

*α* and *β* are coefficients related to growth and non-growth product formation.

The rate of substrate consumption is closely related to the cell’s growth rate and the rate of product formation. This relationship is usually given by the following equation:(8)$$\frac{dS}{dt}=-\frac{1}{Y_{X/S}}\frac{dX}{dt}-\frac{1}{Y_{P/S}}\frac{dP}{dt}-m_{s}X$$
where *Y*_*X*/*S*_ and *Y*_*P*/*S*_ are yield coefficients for biomass on substrate and product on substrate, respectively, g/g, while *m_S_* is the biomass maintenance energy coefficient, g/g.h. In many cases, the latter was neglected.

## 3. Results and Discussion

### 3.1. Isolation and Identification of Lactiplantibacillus plantarum AC 11S

*L. plantarum* AC 11S was isolated from a sample of white brined cheese, homemade on a small farm, in the village of Arda, Rodopa mountain, Bulgaria, near the border with Turkey. The strain is a part of the laboratory collection of the “Stephan Angeloff” Institute of Microbiology, BAS, Bulgaria.

The strain AC 11S was characterized as a Gram-positive, catalase, and oxidase-negative rod-shaped cell morphology (Figure 1a), non-motile, non-spore-forming facultative anaerobe. It was initially identified as a mesophilic bacterium presumptive *Lactiplantibacillus plantarum* by carbohydrate fermentation test with 49 carbon sources (using API 50 CHL, bioMérieux, Marcy l’Etoile, France). The 16S rDNA sequence also showed a similarity with the *L. plantarum* phylogenetic group of the *Lactobacillaceae.* The accurate species affiliation was achieved by multiplex PCR (Figure 1b), according to Torriani et al. [17]. With primers, targeting the *rec A* gene, a PCR product 318 bp was obtained corresponding to the species *L. plantarum.* This molecular method was preferred as a discriminative approach for three highly similar species from the group of *Lactiplantibacillus*—*L. plantarum*, *L. pentosus*, and *L. paraplantarum* as reported by Toriani et al. [17] and Georgieva et al. [18].

### 3.2. Viability of L. plantarum AC 11S in an Acidic Environment

*L. plantarum* is one of the most often employed as a starter LAB species and/or as an adjunct for various biotechnological processes and food production [28]. So, it may be exposed to various stress conditions, such as temperature and osmotic stress freeze-drying in fermented food [29]. In addition, probiotics must withstand extremely high acidity levels and a variety of digestive enzymes throughout the entire gastrointestinal tract [30]. Acid stress is a pivotal issue for microbial survival.

*L. plantarum* AC 11S showed extremely high viability under in vitro passage in simulated stomach juice. After 3 h exposure to pH 1.5 and pepsin, the growth dynamics are similar to the control. The growth was monitored spectrophotometrically (by INNO reader) and the obtained data were summarized (Figure 2). Just a small difference with a longer lag phase was observed. Moreover, the control showed an earlier stationary phase (Figure 2—red line).

### 3.3. Influence of pH and Temperature

Fermentation productivity is strongly influenced by the medium pH value and temperature. Although LAB can grow in a broad range of temperatures and pH levels, their growth rate and population density are affected by these factors [12].

With the aim of determining optimum conditions for *L. plantarum* AC 11S, two series of experiments were carried out. In the first one, the initial medium pH varied from 4.5 to 8.5 at 30 °C, and in the second set of experiments, conducted at pH 6.5, the temperature was changed from 24 to 40 °C. Then, 100 mL LA broth was inoculated with 10% seeding culture and fermentation was carried out at appropriate temperature and pH value under anaerobic and static conditions for 24 h.

The results of these experiments are presented in Figure 3. As can be seen from the figure, the optimum conditions for lactic acid production are pH = 6.5 and temperature of 30 °C. The influence of pH is more pronounced, especially on the cell concentration.

*L. plantarum* is a mesophilic bacterium, capable of growing at temperatures between 10 and 40 °C. In support of our findings, other researchers also cultivated *L. plantarum* at 30 °C as an optimal temperature [31,32,33,34,35], but there are enough investigations in which the strain was cultivated at 37 °C—see for example [5,6,9].

*L. plantarum* possesses high acidity tolerance and can grow in pH values between 4 and 7. It is generally accepted that lactic acid production is a growth-associated process. During the fermentation, due to the lactic acid accumulation, the pH value of the broth decreases to about 3.0–3.5 in pH uncontrolled mode [36,37]. Many authors assumed that both dissociated and undissociated forms of the acid could exhibit an inhibition effect on cell growth, as the undissociated form is the stronger inhibitor [9,13,15]. It is logical because at low pH (below pKa), the acid is mainly in undissociated form. As pointed out by Peetermans et al. [38], other factors besides pKa, like volatility and lipophilicity of the acid, as well as medium pH or acid concentration, influence the microbial growth inhibition in the presence of weak acids. Mercier et al. [39] investigated lactic acid fermentation with glucose as a substrate in the pH range of 5.4–7.8. Based on experimental results, the authors suggested a pH value between 6.0 and 6.5 as optimal for maximal yields for biomass and lactic acid production. W. Fu and A.P. Mathews [9] studied the lactic acid production from lactose with *L. plantarum* in the pH range of 4.0–7.0 and found optimal values for cell growth and acid production between 5.0 and 6.0. Yetiman et al. [4] characterized a new *L. plantarum* strain isolated from *shalgam*—a traditional fermented beverage. The authors investigated the cell growth at different pH values (2, 3, 4, 5, and 7) at two temperatures—30 and 37 °C. Maximum cell density was achieved at pH = 7.0 for both temperatures, but the lag phase was shorter, and the specific growth rate was higher at 37 °C. The same behavior was observed in the presence of different concentrations of bile salts. It is worth mentioning that despite a longer lag phase and lower specific growth rate, the final cell densities were higher at 30 °C, especially in the presence of bile salts.

### 3.4. Influence of Initial Substrate Concentration

In order to elucidate the influence of substrate concentration on cell growth and lactic acid production, a set of experiments was carried out at 30 °C and initial pH 6.5 with lactose monohydrate concentrations of 11, 22, 33, 44, and 55 g/L without pH control during fermentation. The results obtained are presented in Figure 4. From the figure, it is clearly seen that while at 11 g/L substrate concentration, the conversion is almost complete, and close to the theoretical. The degree of conversion decreased with increasing substrate concentration, probably due to the inhibition of high substrate concentration and/or product accumulation. The values of degree of conversion were 100, 76, 51, 39, and 34%, respectively. It is worth mentioning that during the fermentation period (about 50 h), a decrease in the biomass concentration was not observed.

### 3.5. Modeling of Cell Growth, Substrate Consumption, and Product Accumulation

An algorithm for simultaneously solving the model equations describing the fermentation process at different initial substrate concentrations was developed.

Using this algorithm, our experimental data were used to calculate parameters’ values $\left(\mu_{max},K_{S},K_{P},K_{i},X_{max},Y_{X/S},\alpha ,\beta ,Y_{P/S}\right)$ in the model equations.

For this purpose, the experimental data were processed by minimization procedure of the target function *Q*, being the sum of the squares of the differences between the measured biomass, substrate, and lactic acid concentrations and calculated concentrations from the models:(9)$$\begin{array}{l} Q\left(\mu_{max},K_{S},K_{P},K_{i},X_{max},Y_{X/S},\alpha ,\beta ,Y_{P/S}\right)=\\ \frac{1}{N}\overset{N}{\underset{i=1}{\sum }}{\left(X\left(t_{i}\right)-X^{exp}\left(t_{i}\right)\right)}^{2}+\frac{1}{N}\overset{N}{\underset{i=1}{\sum }}{\left(S\left(t_{i}\right)-S^{exp}\left(t_{i}\right)\right)}^{2}+\frac{1}{N}\overset{N}{\underset{i=1}{\sum }}{\left(P\left(t_{i}\right)-P^{exp}\left(t_{i}\right)\right)}^{2} \end{array}$$
where *i* is the experimental point number; *X*, *S*, and *P* are values of biomass, substrate, and product concentration calculated by the model; *X^exp^*, *S^exp^*, and *P^exp^* are experimental values; $t_{i}\left(i=1,\ldots ,N\right)$ are the times in which the biomass (*X*), substrate (*S*), and lactic acid (*P*) concentrations are quantified; and *N* is the experimental data number.

A derivative-free method (fminsearch) is used to find the minimum of the target function Q (least square function) of several variables on an unbounded domain, using MATLAB 2013A software.

As can be seen in Figure 5, no cell death phase was observed during the process. The same holds for other initial substrate concentrations.

Therefore, in Equation (1), the second term was omitted as well as the last term in Equation (4) and the set of differential equations describing the process became
(10)$$\frac{dX}{dt}=\mu X\;\frac{dS}{dt}=-\frac{1}{Y_{X/S}}\frac{dX}{dt}-\frac{1}{Y_{P/S}}\frac{dP}{dt}\;\;\frac{dP}{dt}=\alpha \frac{dX}{dt}+\beta X$$
with initial conditions *t* = 0, *X* = *X*_0_, *S* = *S*_0_, *P* = *P*_0_.

With the aim of finding the best expression for specific biomass growth rate, all kinetic models listed in Table 1, as well as the modified logistic and Gompertz equations, were used.

The attempt to solve the model with the data for all five substrate concentrations was not very successful. The discrepancies between the model and experimental data were large and the Q function value was too big, over 5, and in some cases, the model did not converge.

Analyzing the data for product accumulation in Figure 4, it can be concluded that while at 11 g/L substrate concentration, the conversion is almost complete, increasing the substrate concentration from 22 to 55 g/L leads to strong inhibition of the process and the conversion drops up to about 30%. It was decided to solve the model for 11 g/L separately, using an expression for *μ* without any additional terms for substrate and product inhibition. Four equations were used—the Monod equation, Verhulst equation, and the modified logistic and modified Gompertz equations.

The results are presented in Figure 6 and the values of the model’s parameters are given in Table 2. All four equations described very well the biomass growth, and Gompertz and Verhulst’s equations gave the best fit. Solving together the system for biomass growth, substrate consumption, and product formation, however, the results were a little bit different—the best fit was obtained with the Verhulst equation. At the same time, the value of the Q function was higher in the case of the Gompertz equation and the discrepancies between model and experimental data were high for substrate and product. The obtained values for maximum cell concentration (*Xmax* and *A*) were very close, as well as some other model parameters. A very short lag phase was observed, and the modified Gompertz and logistic models predicted 3–4 h. From the values of the parameters *α* and *β*, it is evident that the production of lactic acid with *L. plantarum* AC 11S is related to biomass growth and there is practically no lactic acid production during the stationary phase.

As for the rest of the experimental data (from 22 to 55 g/L initial substrate concentration), the mathematical model was solved using all the equations listed in Table 1, including different types of inhibition in the terms for specific growth rate *μ*.

Calculated values of model parameters from simultaneous solutions for all four initial substrate concentrations are listed in Table 3 and presented in Figure 7.

One can see that the best fit (lowest Q value) was obtained with the model proposed by Altıok [20]. The models of Verhulst [19] and Aiba [21] also described the growth and production kinetics very well.

None of the two models including substrate inhibition produce good agreement with experimental data. The same observation was made also by Altıok et al. [20]. Åkerberg et al. [23] also reported that the substrate inhibition was very small compared to the product one.

The values of the yield coefficients *Y*_*X*/*S*_ and *Y*_*P*/*S*_ are of the same magnitude, except for those obtained by the Monod–Jerusalimsky model. The ratio between values of the parameters *α* and *β* is also high, which confirms that product formation is growth-related in this concentration range.

Analyzing the literature data, it is obvious that kinetics constant values are not only strain and substrate specific but also depend on other factors like temperature, pH, media composition, etc. Some of the published data for values of maximal growth rate μ are summarized in Table 4.

Because other authors have solved models separately for each initial substrate concentration, similar calculations were made with the proposed model (Equation (10)). Some of the calculated values are presented in Table 5. Experimental values of *α* were calculated from experimental data for biomass and product according to the following equation:(11)$$\alpha =\frac{\left(P_{f}-P_{0}\right)}{\left(X_{f}-X_{0}\right)}$$
where index *f* denotes final and 0 denotes the initial value.

### 3.6. Model Predictions

One of the advantages of the proper mathematical models of a fermentation process is its ability to predict the system behavior in various conditions. With determined model parameters, some calculations were made with a view to predict the influence of the initial biomass concentration, e.g., inoculum volume, starting point of the inhibition, and lactic acid production at high substrate concentrations. To elucidate the influence of biomass on the growth and lactic acid production, the model was solved for four initial biomass concentrations—1, 2, 5, and 8% *v*/*v* inoculum. The results are presented in Figure 8. As can be seen from the figure, an increase in inoculum volume led to a shortening of the lag phase, as well as a reduction in the time necessary to reach the stationary phase. As lactic acid production is growth-associated, an increase in the initial biomass resulted in an increase in lactic acid produced. From the inner figure, it is evident that lactic acid production increased from about 60% to 98% (compared to 10% used in all experiments) when seed volume increased from 1 to 8%.

The model was also solved for six initial substrate concentrations between 11 and 22 g/L. The model predictions are presented in Figure 9. From the inner picture, one can see that the inhibition starts between 15 and 16 g/L.

Finally, the model was solved for higher initial substrate concentrations—from 60 to 100 g/L—and the results are presented in Figure 10. The model predicted a very low increase in product concentration—from 17.07 g/L at 55 g/L to 18.00 at 100 g/L. These values are close to the P_max_ value determined according to model 3 (see Table 3).

## 4. Conclusions

Lactic acid fermentation on lactose by means of *Lactiplantibacillus plantarum* strain AC 11S was investigated in batch mode without pH control. The optimum pH and temperature were determined to be 6.5 and 30 °C. Lactic acid production by *L. plantarum* is growth-associated, and at these conditions, maximum lactic acid production (about 18 g/L) was achieved at 55 g/L initial lactose concentration. The strain is extremely acid-resistant. After 3 h exposure to pH = 1.5 and pepsin, it preserved high viability. A mathematical model was developed including equations describing biomass growth, substrate consumption, and product formations. When solving the model only for bacterial growth, the modified Gompertz equation gave the best fit for all initial substrate concentrations in the investigated range. Including this equation in the system with equations for substrate depletion and product accumulation, however, did not produce a good fit with experimental data. The best fit was achieved with a modified Verhulst equation with an added product inhibition term. Solving the model for initial substrate concentrations from 10 to 20 g/L, it was calculated that the product inhibition starts at 15 g/L. The model results for the influence of initial biomass concentration on bacterial growth showed that an increase in inoculum volume led to a shortening of the lag phase duration and time for reaching the stationary phase. The best results were obtained with 10% inoculum volume. The results obtained are a good base for further investigation of the strain capabilities and the investigations will continue with comparison of model predictions and experimental results in pH-controlled fermentation and at fed-batch and continuous mode.

## Figures and Tables

**Figure 1 microorganisms-12-00739-f001:**
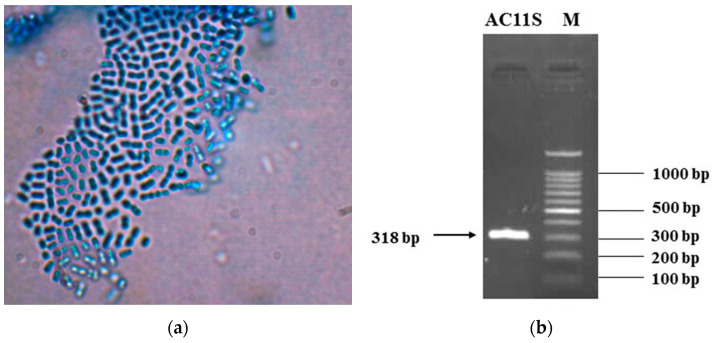
(**a**) Cell morphology of the strain AC 11S (a Gram staining protocol, 1000× dimension light microscopy Boeco, China microscope); (**b**) identification of the strain as *L. plantarum*, with Multiplex PCR using primers targeting *recA* gene, according to Torriani et al. [17].

**Figure 2 microorganisms-12-00739-f002:**
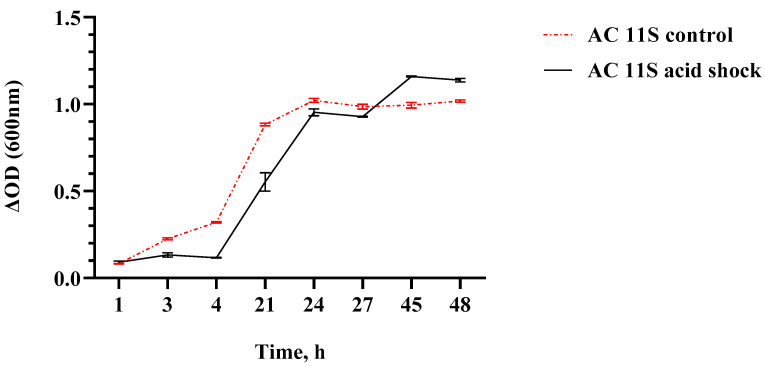
Growth curve of *L. plantarum* AC 11S in mMRS broth with lactose (pH 6.5) at 30 °C after an acidic shock in simulated stomach juice. ΔOD = OD(t_x_) − OD(t_0_), where t_x_ = OD_600nm_ at corresponding time point and t_0_ = OD_600nm_ at 0 h.

**Figure 3 microorganisms-12-00739-f003:**
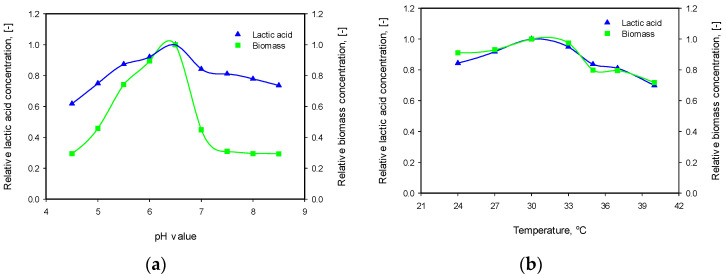
(**a**) Influence of initial pH value on biomass growth and lactic acid production at 30 °C; (**b**) influence of temperature on biomass growth and lactic acid production at initial pH 6.5.

**Figure 4 microorganisms-12-00739-f004:**
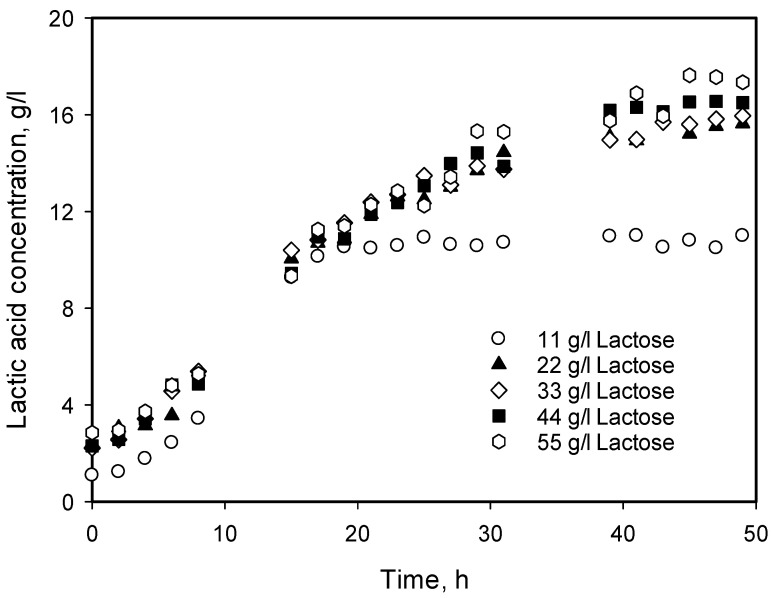
Influence of initial substrate concentration on the lactic acid production at 30 °C and initial pH value 6.5 without pH control.

**Figure 5 microorganisms-12-00739-f005:**
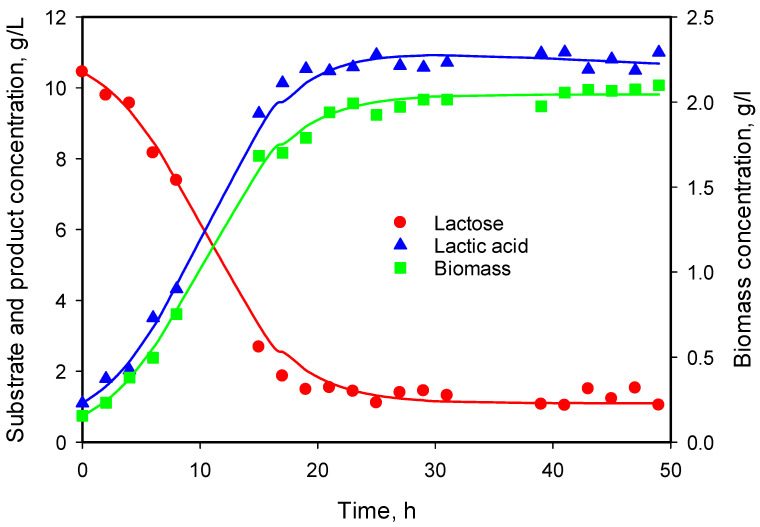
Time course of substrate consumption, biomass growth, and lactic acid production from 11 g/L lactose monohydrate at 30 °C and initial pH value 6.5 without pH control.

**Figure 6 microorganisms-12-00739-f006:**
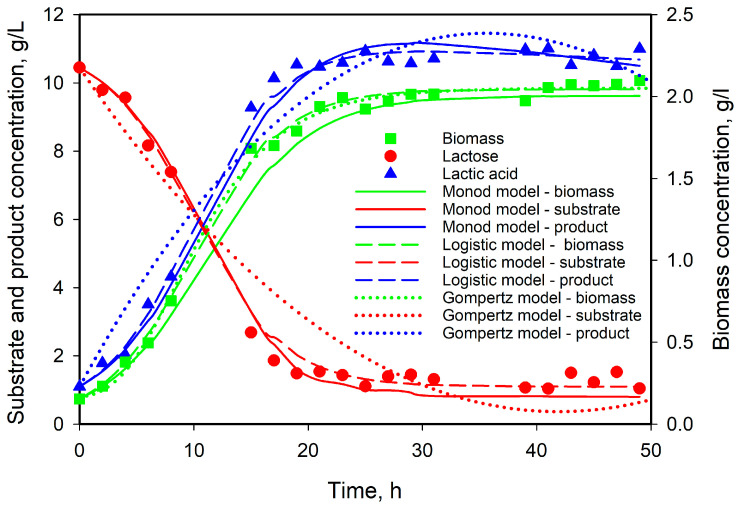
Experimental and model results for microbial growth, substrate consumption, and lactic acid production from 11 g/L initial substrate concentration.

**Figure 7 microorganisms-12-00739-f007:**
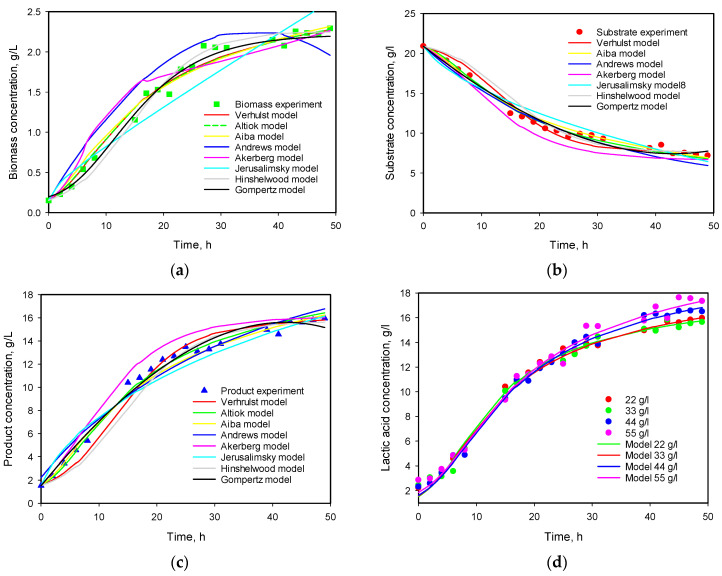
(**a**) Comparison of various model data for biomass growth—22 g/L initial substrate concentration; (**b**) comparison of various model data for substrate consumption—22 g/L initial substrate concentration; (**c**) comparison of various model data for product formation—22 g/L initial substrate concentration; (**d**) comparison of experimental data and Altıok model predictions for lactic acid production—22–55 g/L initial substrate concentration.

**Figure 8 microorganisms-12-00739-f008:**
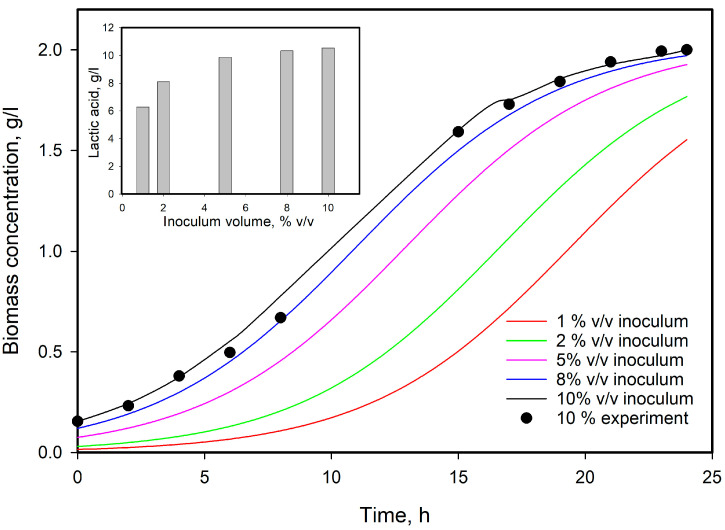
Model predictions by modified Verhulst model for initial biomass concentrations on the cell’s growth and lactic acid production at 11 g/L initial lactose concentration.

**Figure 9 microorganisms-12-00739-f009:**
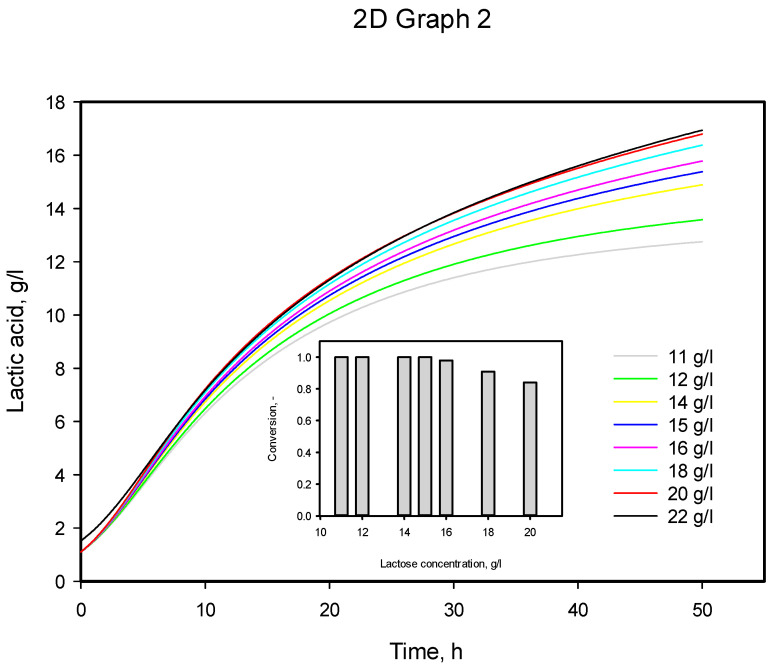
Model predictions by modified Verhulst model for initial substrate concentrations in the range of 11–22 g/L.

**Figure 10 microorganisms-12-00739-f010:**
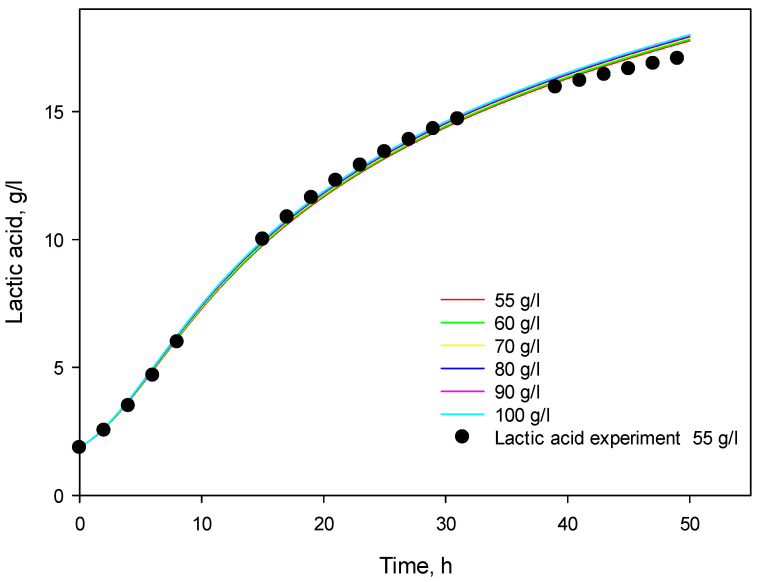
Model predictions by modified Verhulst model for initial substrate concentrations in the range of 55–100 g/L.

**Table 1 microorganisms-12-00739-t001:** Modified Monod models describing cell growth.

No.	Model Name	Equation of Biomass Specific Growth Rate	Reference
1	Verhulst	$\mu =\mu_{max}{\left(1-\frac{X}{X_{max}}\right)}^{n}$ if *n* = 1 $\mu =\mu_{max}\left(1-\frac{X}{X_{max}}\right)$	[19]
2	Altıok	$\mu =\mu_{max}{\left(1-\frac{X}{X_{max}}\right)}^{n_{1}}{\left(1-\frac{P}{P_{max}}\right)}^{n_{2}}$ if *n*_1_, *n*_2_ = 1 $\mu =\mu_{max}\left(1-\frac{X}{X_{max}}\right)\left(1-\frac{P}{P_{max}}\right)$	[20]
3	Aiba	$\mu =\mu_{max}\left(\frac{S}{K_{S}+S}\right)e^{-k_{P}P}$	[21]
4	Andrews	$\mu =\mu_{max}\left(\frac{S}{K_{S}+S+S^{2}/K_{i}}\right)$	[22]
5	Akerberg	$\mu =\mu_{max}\left(\frac{S}{K_{S}+S+S^{2}/K_{i}}\right){\left(1-K_{P}P\right)}^{n}$	[23]
6	Monod–Jerusalimsky	$\mu =\mu_{max}\left(\frac{S}{K_{S}+S}\right)\left(\frac{K_{P}}{K_{P}+P}\right)$	[24]
7	Hinshelwood	$\mu =\mu_{max}\left(\frac{S}{K_{S}+S}\right)\left(1-K_{P}P\right)$	[25]

*μ_max_*—maximum growth rate, h^−1^; *X*—biomass concentration, g/L; *X_max_*—maximum biomass concentration, g/L; *P*—product concentration, g/L; *P_max_*—maximum product concentration, g/L; *S*—substrate concentration, g/L; *K_S_*—the substrate saturation constant, g/L; *K_P_*—product formation constant, g/L; *K_i_*—product inhibition constant, g/L.

**Table 2 microorganisms-12-00739-t002:** Model parameters determined by solving the system 10 for 11 g/L initial substrate concentrations.

Model	*μ_max_*	*K_S_*	*X_max_*	*λ*	*A*	*Y* _*X*/*S*_	*Y* _*P*/*S*_	*α*	*β*	*Q*
Monod	0.420	5.511				0.175	4.016	5.965	0.0197	1.019
Verhulst	0.253		2.045			0.202	6.394	5.396	0.0077	0.352
Gompertz	0.048			2.959	1.925	0.205	5.986	6.086	0.0042	3.846
Logistic	0.138			4.0705	1.875	0.229	2.011	4.591	0.0	1.949

*μ_max_*—maximum growth rate, h^−1^; *K_S_*—substrate saturation constant, g/L; *X_max_*—maximum biomass concentration, g/L; *A*—the asymptotic level of biomass, g/L; *λ*—duration of the lag phase, h; *Y*_*X*/*S*_ and *Y*_*P*/*S*_—yield coefficients for biomass on substrate and product on substrate, respectively, g/g; *α* [-] and *β* [h^−1^]—growth-associated constants for product formation and the non-growth-associated constant for product formation, h^−1^; *Q*—target function.

**Table 3 microorganisms-12-00739-t003:** Model parameters determined by solving the system 10 for 22–55 g/L initial substrate concentrations.

Model No	*μ_max_*	*K_S_*	*K_P_*	*K_i_*	*X_max_*	*P_max_*	*n* _1_	*n* _2_	*Y* _*X*/*S*_	*Y* _*P*/*S*_	*α*	*β*	*Q*
1	0.179				1.754				0.121	3.650	8.337	0.0156	0.991
	0.322				3.673		2.060		0.154	3.346	7.015	0.0014	0.348
2	0.257				2.268	19.389			0.419	3.803	7.263	0.0069	0.494
	0.302				3.013	17.087	2.248	1.767	0.151	2.951	6.948	0.0029	0.305
3	0.365	3.464	0.247						0.155	1.855	6.784	0.005	0.463
4	0.168	0.550		0.077					0.144	5.269	6.232	0.0549	4.099
5	0.113	17.306	2.506	6.406					0.011	58.27	29.07	1.2367	4.71
6	0.185	4.293	1.052						0.114	1.231	7.287	0.0231	3.287
7	0.233	4.847	0.064						0.158	3.262	6.524	0.0028	1.348

*μ_max_*—maximum growth rate, h^−1^; *K_S_*—substrate saturation constant, g/L; *K_P_*—product formation constant, g/L; *K_i_*—product inhibition constant, g/L; *X_max_*—maximum biomass concentration, g/L; *P_max_*—maximum product concentration, g/L; *n*_1_, *n*_2_—coefficients, *Y*_*X*/*S*_ and *Y*_*P*/*S*_—yield coefficients for biomass on substrate and product on substrate, respectively, g/g; *α* [-] and *β* [h^−1^]—growth-associated constant for product formation and the non-growth-associated constant for product formation, h^−1^, *Q*—target function.

**Table 4 microorganisms-12-00739-t004:** Values of the maximal growth rate determined by other authors.

Microorganism	Substrate	Growth Model	*μ_max_* (h^−1^)	Reference
*L. amylophilus*	Glucose 20 g/L	$\mu =\mu_{max}\;\left(1-X⁄X_{max}\right)$	0.32 (pH 6.5)	[39]
*L. casei*	Lactose 20 g/L	$\mu =\mu_{max}\;\left(1-X⁄X_{max}\right)$	0.511	[40]
*L. plantarum*	Glucose 100 g/L	$\mu =\mu_{max}\;\left(1-X⁄X_{max}\right)$	0.64	[41]
*L. lactis*	Glucose 20 g/L	$\mu =\left(\mu_{max}S\left(K_{P}+P\right)\right)/\left(\left(K_{S}+S\right)P\right)$	0.66 (pH 6.5)	[13]
*L. acidophilus*	Glucose 10 g/L	Four-parameter Gompertz model	0.35 (pH 6.5, 30 °C)	[42]
			0.43 (pH 6.5, 37 °C)	
*L. helveticus*	Whey ultrafiltrate powder	$\mu =\mu_{max}1/\left(1+ce^{dt}/\left(\mu^{m}-c\right)\right)$	0.56	[43]
*L. paracasei*	Rice flour	$\mu =\mu_{max}\left(S/\left(S+K_{S}X\right)\right)$ $\mu =\mu_{max}\left(1-X/X_{max}\right)$ $\mu =\mu_{max}S/\left(K_{S}+S\right)$	0.993	[44]
			0.619	
			0.811	
*L. plantarum*	Sucrose 20 g/L	$\mu =\mu_{max}\;\left(1-{\left(X⁄X_{max}\right)}^{n}\right)$.	0.0545	[45]
*L. plantarum*	Hydrolyzed wheat flour	$\mu =\mu_{max}\left(S/\left(K_{S}+S+S^{2}/K_{i}\right)\right){\left(1-K_{P}P\right)}^{n}$	0.403	[23]
*L. plantarum*	Lactose 40 g/L	$\mu =\mu_{max}S/\left(K_{S}+S\right)$	0.364 (pH 6.0)	[9]
*L. plantarum*	Dairy waste water	$\mu =\mu_{max}\;\left(1-X⁄X_{max}\right)$	0.35	[6]
*L. casei*	Waste potato starch	$\mu =\mu_{max}S/\left(K_{S}+S\right)$	0.115	[46]
*L. lactis*	Glucose	$\mu =\mu_{max}\;\left(1-X⁄X_{max}\right)$	0.687	[47]
*L. amylovorus*	Glucose	$\mu =\left(\mu_{max}S/\left(K_{S}+S\right)\right)\left(K_{I}/\left(K_{I}+P\right)\right)$	0.58	[48]
	Sucrose		0.32	
	Starch		0.61	
*L. delbrueckii*	Glucose 10 g/L	Logistic equation	0.031	[49]
*L. delbrueckii*	Glucose	$\mu =\mu_{max}/\left(1+S/K_{I}\right)$ $\mu =\mu_{max}e^{-S/K_{I}}$ $\mu =\mu_{max}\;{\left(1-S/S_{m}\right)}^{n}$	0.55	[50]
			0.59	
			0.58	
*L. casei*	Whey lactose	$\mu =\mu_{max}\;\left(S/\left(K_{S}+S\right)\right)\left(1-X⁄X_{max}\right)$	0.265	[20]
*L. delbrueckii*	Potato starch	$\mu =\mu_{max}e^{-K_{P}P}\left(K_{S}+S+S^{2}/K_{i}\right)$	0.372 (pH 5.5)	[51]
*L. helveticus*	Whey lactose	$\mu =\mu_{max}\;\left(1-X⁄X_{max}\right)$	0.64	[52]
*L. helveticus*	Lactose 50 g/L	$\mu =\left(\mu_{max}S/\left(K_{S}+S\right)\right)e^{-{\left(S/K_{I}\right)}^{n1}}e^{-{\left(P/K_{P}\right)}^{n2}}$	0.25	[53]
*L. casei*	Glucose 50 g/L	$\mu =\mu_{max}\left(K_{P}/\left(K_{P}+P\right)\right)\left(S/\left(K_{S}+S\right)\right)\left(1-P/P_{c}\right)$	0.45	[54]
*L. plantarum*	Irish brown seaweeds	Modified Gompertz equation		[55]
	*L. digitata*		0.4	
	*L. saccharina*		0.29	
*L. plantarum*	Glucose—20 g/L Vegetable juice—3 g/L RS	Modified Gompertz equation	0.6	[56]
			0.45	
*L. plantarum*	Glucose—20 g/L Vegetable juice—3 g/L RS	Logistic equation	0.62	[56]
			0.53	

*μ_max_*—maximum growth rate, h^−1^; *S*—substrate concentration, g/L; *S_m_*—maximum substrate concentration, g/L; *K_S_*—substrate saturation constant, g/L; *K_P_*—product formation constant, g/L; *K_i_*—product inhibition constant, g/L; *X*—biomass concentration, g/L; *X_max_*—maximum biomass concentration, g/L; *P*—product concentration, g/L, *P_c_*—critical product concentration, g/L; *A*—the asymptotic level of biomass, g/L; *λ*—the duration of the lag phase, h; *Y*_*X*/*S*_ and *Y*_*P*/*S*_—yield coefficients for biomass on substrate and product on substrate, respectively, g/g; *α* [-] and *β* [h^−1^]—growth-associated constants for product formation and the non-growth-associated constant for product formation, h^−1^.

**Table 5 microorganisms-12-00739-t005:** Experimental and calculated values of *α* and *µ_max_*.

Initial Substrate Concentration, g/L	*α exp*	*α* calc	*µ_max_* _, calc_
11	5.091	6.850	0.249
22	6.748	6.782	0.334
33	6.959	6.768	0.343
44	6.947	7.197	0.311
55	7.651	7.195	0.309
Mean value	6.679	6.958	0.309

Both mean values are very close to the value predicted by the model solving all data for different substrate concentrations simultaneously.

## Data Availability

Not applicable.

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
