# Peer review of "Lactic Acid Production by Lactiplantibacillus plantarum AC 11S—Kinetics and Modeling"

_microorganisms, 2024, doi:10.3390/microorganisms12040739_

Round 1

Reviewer 1 Report

Comments and Suggestions for Authors

The text presents a comprehensive examination of the conversion of lactose by Lactiplantibacillus plantarum AC 11S. The study delves into crucial factors such as initial substrate concentration, optimal pH, and temperature, offering practical insights for potential industrial applications. By identifying the optimal pH (6.5) and temperature (30°C) for bacterial growth and lactic acid production, the findings gain practical significance.The study employs a thoughtful approach by comparing various models, emphasizing their strengths and limitations. Notably, the selection of the modified Gompertz equation for biomass growth and a variant of the logistic equation for product inhibition reflects a well-considered choice in modeling. However, it is essential to note a potential limitation in the experimental setup. The investigation is carried out without maintaining a constant pH, introducing variability to the conditions. Given that the optimum pH for lactic acid fermentation is determined to be 6.5, the absence of pH control may result in fluctuations during the fermentation process. This uncontrolled variable has the potential to impact the reproducibility and reliability of the study's results.

Author Response

Thank you very much for your time and efforts spent reviewing our manuscript.

We agree that conducting the fermentation without pH control has some drawbacks and limitations. This investigation aimed to determine the optimum conditions and to choose the best model describing the process of lactic acid production from lactose. We have experimental data for lactic acid production at constant pH. The modeling results showed differences in model parameters’ values for the two fermentation modes. These results and comparisons with uncontrolled pH fermentation will be subject to our next publication.

Reviewer 2 Report

Comments and Suggestions for Authors

The article entitled "Lactic acid production by Lactiplantibacillus plantarum AC 11S – kinetics, and modeling" investigates lactic acid production from lactose by Lactiplantibacillus plantarum AC 11S, focusing on the kinetics and modeling aspects.

I find the Introduction well-written! 

Lines 86, 99: replace "Lactiplantibacillus plantarum" with L. plantarum

Some equations are not entirely visible (readable).

The discussions are well writen and compared with the literature. 

Please develop the conclusions (mention specific results) and discuss something about future directions. 

Author Response

Thank you very much for your time and efforts spent reviewing our manuscript.

Comment 1: Lines 86, 99: replace "Lactiplantibacillus plantarum" with L. plantarum.

Response 1: Proposed changes were made.

Comment 2: Some equations are not entirely visible (readable).

Response 2: We are very sorry for this, but in our opinion, this is due to the conversion of the manuscript to PDF format. In the original uploaded MS Word file, all equations are correct and readable.

Comment 3: Please develop the conclusions (mention specific results) and discuss something about future directions.

Response 3: The conclusions were broadened by adding some new data.

Reviewer 3 Report

Comments and Suggestions for Authors

The aim of this manuscript was to find the optimal conditions of lactic acid production by a Lactiplantibacillus plantarum stain using mathematical models. A specific issue is to better describe how these equations have been formulated, and the meaning of the obtained values. The introduction is quite general, and should include the mathematical reasonings. The biological experiments are quite simple, and should also include changes in initial bacterial densities.

·         The abstract lacks a background sentence, hypothesis, and the reason for doing this research,

·         Line 16: a comma should be added after "associated" since the second part of the sentence is not connected with the first part. The growth-associated relationship should be better described in the abstract.

·         Line 16: Are you sure that you mean "substrate concentration" and not "lactic acid concentration"?

·         The last sentence of the abstract is long with synthetic errors. It needs to be divided into two-three sentences.

·         A conclusion should be added to the abstract.

·         Line 26: Delete the comma.

·         Line 27: Instead of "Their main", I would suggest writing "one of their".

·         Line 29: A comma should be added after "anaerobic".

·         Line 32: Please remove comma after (EPS).

·         Line 33: an overwritten comma should be deleted.

·         Line 36: correct to "meat".

·         Line 49: Define the difference between heterofermentative and homofermentative.

·         Line 50: The authors wrote: ", it shares different metabolic features". These features should be described in the text.

·         Line 52: Please set "lactobacilli" in italics.

·         Line 60: Delete comma after "fatty".

·         Line 81: It is written "a promising producer of lactic acid": Please describe in the introduction why the production of lactic acid is important.

·         Line 88: Delete comma after "broth".

·         Line 90: correct to: "in growth medium containing".

·         Lines 90-92: The concentrations can be added before the name of the components.

·         Line 92: Some of the numbers should be set in subscript.

·         Line 95: an "L" is lacking in mL.

·         Line 109 should be in past tense.

·         The assay determining lactic acid production should be better described.

·         Pages 4, 5, 9: The formulas have Chinese letters. Please rewrite.

·         Line 116: Describe how the pH was adjusted to pH 1.5, and how the pH was altered during the bacterial growth.

·         Line 147-etc: You can't say "concentration" about cells. It should be "bacterial density" – maybe you mean "biomass" here? Describe how the specific growth rate is defined. It may differ from lag, log and stationary phase.

·         How was the cell death rate determined?

·         The Method section should include a statistical section.

·         Line 183: There is an extra underline that should be deleted.

·         Line 193: Is it sufficient to determine the PCR size, or the PCR product needs to be sequenced?

·         Figure 1: in Legend A, describe that this is Gram staining.

·         Line 200: Correct to "gene" and add a closing parenthesis.

·         Line 211: I think you can delete "may be pointed out".

·         The time of incubation should be added to the figure legend.

·         Figure 2: The OD is presented as delta. This needs explanation. Also, the OD is here 600nm, which is in contrast to the 620nm described in the Method section.

·         Figure 3: How can g/l be relative?

·         Lines 235-238 belong to Discussion.

·         Also, the next paragraph (Lines 239-261) belongs to Discussion or can be described in Introduction. The Result section should only deal with your experiments and data observed.

·         Line 240: There is twice the word "growth".

·         "Hours" should be "h" throughout the text.

·         Line 268: "The degree of conversion" should be calculated.

·         Line 271: Cell death can not be determined by OD, but by CFU. The bacteria didn't reach the "dead phase" where cell death exceeds bacterial growth. Ths the sentences should be rephrased.

·         Section 3.5 should initiate with a description of what was the question, and then present the data.

·         Figure 5: How was lactose determined?

·         The biomass was determined by OD. How can you convert the OD to g/L?

·         Line 306: I am not sure you can say "inhibition" here. Maybe you reached the highest bacterial capacity to produce lactic acid.

·         Table 2: The meaning of each parameter should be described.

·         Spelling mistake in Figure 7b: Please correct to "Jerusalimsky".

·         Y-axis in Figure 7a: As asked above – how did you convert the OD to biomass concentration?

·         Line 348: You can't use the word "both" here.

·         Could the inhibition be due to product inhibition – maximum level of tolerable lactic acid.

·         Line 329: Please better phrase the sentence.

·         Lines 356-359 and Table 4 belong to Discussion.

·         In Table 4 there are a lot of question marks. What is the intention of the question marks?

·         The different parameters in Table 4 should be defined.

·         Line 375: Again, the question is whether it is "inhibition" or you reached the lactic acid concentration that inhibited further lactic acid production.

·         The paper lacks a Discussion Section.

·         Line 389: If the strain is acid resistance, the text saying "easily recovers from an acid shock" has no value.

·         The conclusion should also include the impact of biomass and bacterial density on lactic acid production.

·         Also, the message of this manuscript should be better emphasized in the conclusion section.

Comments on the Quality of English Language

Only minor English corrections needed.

Author Response

Thank you very much for your time and efforts spent reviewing our manuscript. The thoroughgoing and earnest review, comments, and suggestions will be very helpful in improving the manuscript.

Comment 1: Line 16: a comma should be added after "associated" since the second part of the sentence is not connected with the first part.

Response 1: In lines 16, 26, 27, 29, etc. all comments on grammatical and spelling errors were accepted and corrected.

Comment 2: Line 16: Are you sure that you mean "substrate concentration" and not "lactic acid concentration"?

Response 2: Yes, we are sure, see Fig. 4 and Fig. 8.

Comment 3: The last sentence of the abstract is long with synthetic errors. It needs to be divided into two-three sentences.

Response 3: The sentence was modified as follows: The modified Gompertz equation gave the best fit when solving only the equation for biomass growth at different initial substrate concentrations. Solving the entire set of differential equations for bacterial growth, substrate consumption, and product formation, the best results were obtained when using a variant of the logistic equation for biomass growth. This variant included a term for product inhibition and described in the best way all experimental data.

Comment 4: Line 49: Define the difference between heterofermentative and homofermentative.

Line 50: The authors wrote: ", it shares different metabolic features". These features should be described in the text.

Response 4: The following text was added: The main difference between homo- and hetero-fermentative lactobacilli is the existing difference in the metabolic pathways leading to variation in the main end-product of the fermentation. While homofermentative lactobacilli produce lactic acid as the sole product, the heterofermentative LABs produce a mixture of lactic acid, ethanol, acetic acid, and carbon dioxide during fermentation. Other differences exist in the ability of various LABs to decompose macromolecular substances (polysaccharides and proteins), as well as in the production of bacteriocins, short-chain fatty acids, amines, etc.

Comment 5: Line 81: It is written "a promising producer of lactic acid": Please describe in the introduction why the production of lactic acid is important.

Response 5: The following sentences were added. LABs have been used by humans for centuries. Several metabolic end-products of lactic acid fermentation have practical applications. Lactic acid is a very useful chemical with numerous applications in pharmaceutical, food, cosmetic, and other industries. In recent years the use of LABs in drug delivery systems, fermented foods, and beverages, as well as probiotics and vaccines has gained more and more interest.

Comment 6: Pages 4, 5, 9: The formulas have Chinese letters. Please rewrite.

In Table 4 there are a lot of question marks. What is the intention of the question marks?

Response 6: We are very sorry for this, but in our opinion, this is due to the conversion of the manuscript to PDF format. In the original uploaded MS Word file, all equations are correct and readable.

Comment 7: Line 116: Describe how the pH was adjusted to pH 1.5, and how the pH was altered during the bacterial growth.

Response 7: The pH values were measured using pH meter HANNA instruments HI2211. Correction of pH to pH=1.5 was done with hydrochloric acid. The Lactobacillus cells were incubated for 3 h at 30oC. In such conditions, there was no growth, and no lactic acid production, thus, a significant change in pH has not been expected. However, the cells remained viable. Corresponding text was added to the manuscript.

Comment 8: Line 147-etc: You can't say "concentration" about cells. It should be "bacterial density" – maybe you mean "biomass" here? Describe how the specific growth rate is defined. It may differ from lag, log and stationary phase.

How was the cell death rate determined?

Response 8: Cell concentration was replaced by biomass concentration. The specific growth rate is always defined as µ=dX/Xodt. In the manuscript, many equations for specific growth rate were described and tested. We did not determine the death rate; our model didn’t include a term for cell death.

Comment 9: Figure 1: in Legend A, describe that this is Gram staining.

Figure 2: The OD is presented as delta. This needs explanation. Also, the OD is here 600nm, which is in contrast to the 620nm described in the Method section.

Figure 3: How can g/l be relative?

Figure 5: How was lactose determined?

The biomass was determined by OD. How can you convert the OD to g/L?

Y-axis in Figure 7a: As asked above – how did you convert the OD to biomass concentration? Spelling mistake in Figure 7b: Please correct to "Jerusalimsky".

Response 9: Gram staining was added in the figure captions.

A text explaining delta OD was added.

The spectrophotometric measurement of bacterial growth is usually performed around 600 nm. The difference in measurement (600 and 620 nm) came from the different standardized protocols the two laboratories have used, depending on the spectrophotometer used. As a microbiology laboratory, we monitor the dynamics of bacterial growth by means of OD at 600 nm and/or using the Kоch method, with decimal dilutions reporting CFU/ml.

I apologize for my blunder; the initial figure was in real biomass concentration. Corrections in the figures were made.

Lactose concentrations were measured in the same way as lactic acid – by the HPLC method described.

Biomass concentrations in g/l were calculated from readings of OD using a previously prepared calibration curve for the specific strain.

The spelling mistake in Fig. 7 was corrected.

Comment 9: Line 193: Is it sufficient to determine the PCR size, or the PCR product need to be sequenced?

Response 9: The multiplex PCR using primers targeting the recA gene is more highly discriminative for the 3 closely related species L. plantarum, L. pentosus, and L. paraplantarum (Torriani et al. 2001). The species-specific primers did not work properly. This was confirmed by our publication Georgieva et al. 2008. We successfully applied to identify several L. plantarum strains. The problem with their identification (respectively discrimination of the 3 species cited above) is that they share 99.6% of the 16S rRNA gene similarity. i.e. the sequence analysis by the rrn operon would not be a successful approach. Nevertheless, we have sequenced the 16 rDNA and it confirms the identification of the strain as L. plantarum.

Comment 9: The paper lacks a Discussion Section.

Lines 235-238 belong to Discussion.

Also, the next paragraph (Lines 239-261) belongs to Discussion or can be described in Introduction.

Response 9: According to the Instructions for authors, having separate sections Results and Discussion is not mandatory. It is clearly stated that these sections may be combined and that is the approach authors have chosen.

Comment 10: Line 329: Please better phrase the sentence.

Response 10: The sentence was rewritten.

As for the rest of the experimental data (from 22 to 55 g/L initial substrate concentration) the mathematical model was solved using all the equations listed in Table 1, including different types of inhibition in the terms for specific growth rate μ.

Comment 11: Table 2: The meaning of each parameter should be described.

The different parameters in Table 4 should be defined.

Response 11: Done.

Comment 12: Line 268: "The degree of conversion" should be calculated.

Response 12: The degree of conversion was calculated and added to the text.

Comment 13: Section 3.5 should initiate with a description of what was the question, and then present the data.

Response 13: The text in the section was rearranged.

Comment 14: Line 271: Cell death can not be determined by OD, but by CFU. The bacteria didn't reach the "dead phase" where cell death exceeds bacterial growth. Ths the sentences should be rephrased.

Response 14: The sentence was rephrased.

Comment 15: Line 306: I am not sure you can say "inhibition" here. Maybe you reached the highest bacterial capacity to produce lactic acid.

Could the inhibition be due to product inhibition – maximum level of tolerable lactic acid

Line 375: Again, the question is whether it is "inhibition" or you reached the lactic acid concentration that inhibited further lactic acid production.

Response 15: We believe that in this case, it is inhibition. The decrease in conversion starts at about 20 g/L substrate concentration and with increasing substrate concentration more lactic acid was produced. What is more, the predicted value of maximum product concentration was higher than the experimentally determined. Solving the model with an initial substrate concentration of up to 100 g/l the estimated quantity of lactic acid also increased.

Comment 16: The conclusion should also include the impact of biomass and bacterial density on lactic acid production.

Response 16: All our experiments were done with the same initial biomass concentration, so we didn’t have data for the influence of bacterial density on lactic acid production. However, such influence could be determined by solving the model for different values for biomass. A new figure was added.

Round 2

Reviewer 3 Report

Comments and Suggestions for Authors

Comments to revised version. Popova et al.

There are some improvements in the text, but it is still difficult to follow the models without deeper descriptions (e.g., why these were chosen, what is the reasoning of the new algorithm, and what is the meaning of the different numbers.) This should be much better described. The definition of all parameters should be provided and please describe what can we conclude from the obtained data. Also provide a name to your algorithm.

The Title should be rewritten to present the content. For instance: "Mathematical modeling to predict the kinetics of lactic acid production by Lactiplantibacillus plantarum AC 11S".

Abstract

The abstract should include why it was important to study lactic acid production and what is the aim of this paper (e.g., developing an algorithm to predict lactic acid production by Lactiplantibacillus plantarum AC 11S).

The abstract should have a conclusion.

Spelling mistake in line 23. Please correct to "biomass". Do you mean "shorter" instead of "short".

Methods

Line 111: correct to: "MgSO4·7H2O". and "MnSO4·7H2O"

Line 135: Please correct to "pH 1.5, 0.8% NaCl (w/v) and 3 mg/mL pepsin". The source of the pepsin should be stated.

Line 147: Please correct to " Perkin-Elmer".

Pages 4, 6, 10: In the current pdf – the formula are still with Chinese letters. Please find a way to solve the problem.

You need to add in the text that you have sequenced the 16 rDNA.

The specific growth rate period is defined as the rate of increase of biomass of a cell population per unit of biomass concentration. Thus, the unit of biomass should be defined. Also describe how these parameters were determined.  The same for μmax. How was this determined experimentally? I believe this is a function of the growth stage. Definition of "the substrate saturation constant" should be provided. (e.g., " the concentration of growth rate-limiting nutrient that supports half the maximum specific growth rate).

You stated that: "In many cases, the cell death was not observed, and the second term can be omitted". How did you detect the death rate?  When reaching the stationary phase μ=kd.

References should be provided where you mention formula (e.g. in line 171 for Monad equation; line 181 for logistic equation and Gompertz model).

A reference should be added to the statement: " that lactic acid fermentation is limited by substrate and inhibited by the product".

Table 1: All of the parameters/symbols should be defined.

Results (the word Discussion has been lost).

Line 235: the Delta should be in larger letter and the misspelling correct to "different" I think it would be better to write ΔOD=OD(tx)-OD(t0) and then define tx and t0.

Line 298: Spelling mistake. Correct to: "decrease in".

Line 306: How the MatLab was used to determine the model should be described. The target function Q should be defined here. Also the 7 parameters in parenthesis should be defined, and what is the meaning of the numbers obtained from each parameter. What do you mean by →min. Also, all symbols should be defined. Describe what is the difference between Xj and Xj exp and Sj and Sjexp. i is defined, but it does not appear in the formula.

Line 313: It says: " As can be seen in Figure 5 (growth, substrate, and product for 11 g/L) no cell death was observed during the process". This can not been seen – as the stationary phase is an equilibrium between growth rate and death rate. Therefore, the further text is not accurate, and adjustment of the calculations has to be done.

Please add references after the following sentence: "Many authors assumed that both dissociated and undissociated forms of the acid could exhibit an inhibition effect on cell growth, as the undissociated form is the stronger inhibitor".

Line 339: The results of the data described in sentence beginning with: " The attempt to solve the model" should be presented."

Still the sentence: " while increasing the substrate concentration from 22 to 55 g/L leads to strong inhibition of the process and the conversion drops up to about 30%". The sentence indicates that higher concentrations of lactose inhibits lactic acid production, but it also could be that a maximum lactic acid production has reached due to the inhibition of lactic acid on its own production. Thus, both options should be discussed.

Table 3: The definition of the 7 models should be defined.

Figure 6: The order of Monod (biomass, substrate, product) should be (substrate, product, biomass) in order to be consistent with the others.

Line 368: Correct to: "formation".

Table 4 has still question marks – please correct. K in capital or small letters? Describe how do you calculate substrate saturation constant, critical product concentration, the asymptotical level etc.

In formula 10 (line 424) – which time point was alpha calculated? The first P and X value should be labeled with t= the time point.

Line 432: Correct to "volume"

Line 434: Correct to: "growth"

Line 450: Correct to: "Finally".

Table 2: Each model provides a different number. So how can this be interpreted. Which model is closest to the real data?

Figure 1: The X-axis is not on a scientific scale.

Figure 7: the a-d labels in the current presentation is not clear. Please show only the revised figure, without mixing it with the old figure. If the "product" is "lactic acid", what is the difference between c and d?

Figure 8: What is the difference between 10% experiment and 10% v/v inoculum? The model used should be mentioned. The same for Figures 9 and 10.

Figure 9: why are the numbers in the legend not in increasing order?

When you write "model prediction" – you have to clarify that it is your algorithm.

Line 460. Bacterial name in italics.

Line 461: You can rather say "maximum lactic acid production was achieved at 18 g/L lactose".

The conclusion should add a sentence for recommendation of the optimal conditions (including initial density) for best lactic acid production.

First sentence in Author contribution as well as " The following statements should be used" should be deleted.

Comments on the Quality of English Language

In general the English is OK. Some scientific styling is required.

Author Response

The authors are extremely grateful то the reviewer for the time spent on a thorough review and for providing this valuable feedback.
